# Covariation of Passive–Active Microwave Measurements over Vegetated Surfaces: Case Studies at L-Band Passive and L-, C- and X-Band Active

**Erica Albanesi [1], Silvia Bernoldi [1], Fabio Dell'Acqua [1,*] and Dara Entekhabi [2]**

[1] CNIT, Pavia Unit, Department of Electrical, Computer, Biomedical Engineering, University of Pavia, I-27100 Pavia, Italy; erica.albanesi01@universitadipavia.it (E.A.); silvia.bernoldi01@universitadipavia.it (S.B.)

[2] Department of Earth, Atmospheric and Planetary Sciences, Massachusetts Institute of Technology, Cambridge, MA 02139, USA; darae@mit.edu

\* Correspondence: fabio.dellacqua@unipv.it

**Abstract:** The analysis of soil and land cover scattering properties and their connection with the parameters of microwave scattering is a longstanding research topic. Recently, the advent of modern space-borne microwave radiometers like SMAP in addition to the trend towards open data for scientific use fostered the development of enhanced models based on data fusion from different platforms permitting more accurate assessments. SMAP was designed to operate on an integrated combination of a radiometer and a radar, both operating in L-band. Unexpected failure of the radar component encouraged scientists to experiment various combination of data from the surviving radiometer with other sources of radar data, notably C-band Sentinel-1 data. In this work, we present a case study on a possible combination of SMAP radiometer data with X-band radar data from TerraSAR-X and COSMO-SkyMed, comparing results with those provided by NASA from their standard production procedures. The study was performed on two test sites, one at an agricultural site in Germany and one in the Brazilian Amazon, to explore very different vegetation conditions. This work is a part of a broader research effort addressing the combination of multiple sources of passive and active microwave sensing data. The research question defining this research effort is whether the use of data from multiple active sources affords either obtaining more accurate estimates of active–passive co-variation parameters for a given observation period, or shortening the minimum observation period by increasing the temporal density of active samples. In this framework, this paper addresses a preliminary comparison of fresh and past results obtained from C-, X-, and L-band active sensing data. The observed relations offer interesting clues on the impact of band selection on soil vegetation analysis.

**Keywords:** SMAP; active–passive; microwave sensing; spaceborne sensors; vegetation; covariation parameter

## 1. Introduction

In the context of active–passive microwave sensing of the Earth, signals derived from different sensors can be jointly analyzed by means of a covariation model [1]. When observing a common area of interest, time series of concurrent active and passive microwave signals manifest a mutual variation according to their respective sensitivities to certain physical properties of the region under study. By examining the covariation patterns of data from different sensors observing the same target, a collective data analysis can be performed, which unlocks information about the target that is not accessible through any individual sensor. Fusing active and passive microwave sensor data makes deeper insights accessible about the physical status of the observed surfaces [2]. This is especially appealing in space-based Earth observation, where generally no single type of sensor is suitable to extract all of the variables of interest in understanding a certain phenomenon. In this

context, fusing passive and active microwave sensor data offers the additional advantage of obtaining output data possessing the highest among the spatial resolution levels of the fused sources. The cooperative usage of active and passive microwave signals allows an improved retrieval of specific parameters in an observed region, such as soil moisture or vegetation water content.

Soil moisture, for example, represents a key piece of information for several applications concerning environmental observation [3]. Weather forecasting, agricultural productivity, water resources management, and drought prediction are just some examples among the applications that require detailed knowledge about the status of soil moisture [4]. In addition, water contained in the surface layer of the Earth constitutes a linking factor between water, energy and carbon fluxes through land and atmosphere. Through bare soil evaporation and plant transpiration (evapotranspiration), around 60% of precipitation returns to the atmosphere [5]. Moreover, evapotranspiration consumes more than half of the total solar energy absorbed by land surfaces [6]. Furthermore, a variation in the amount of water stored in land surfaces, which can be caused by climate warming, may change the ecosystem carbon fluxes [7]. Thus, the study of soil moisture provides a methodology for monitoring biosphere conditions and climate equilibrium.

Vegetation water content (VWC) is another important parameter for environmental studies that can greatly benefit from joint active–passive microwave sensing of the Earth surface [8,9], along with vegetation roughness [10]. Global maps of soil moisture and vegetation characteristics are indeed produced relying on data from passive microwave sensors missions like ESA's SMOS [11,12] and NASA's soil moisture active passive (SMAP) mission [13], each coming with its own specific techniques and algorithms [14,15] for retrieval of land cover parameters [16]. In this paper we use data from the SMAP observatory, which was launched in January 2015 and started operating in April 2015. As the name of the mission suggests, the satellite carries an active instrument (radar) and a passive one (radiometer), both operating on the L-band, designed to collectively measure the land surface soil moisture and the freeze/thaw state of the ground in colder regions. The radiometer is able to retrieve the data of interest with high accuracy under the presence of vegetation coverage [17], although only at a coarse spatial resolution (38 km × 49 km). The radar component can produce data at a higher spatial resolution. By merging the L-band radiometer and radar retrievals [13], intermediate resolution and intermediate accuracy soil moisture estimates are achievable. Under the assumption of frequent, concurrent and co-located active and passive measurements, the covariation between the two measurements can be estimated using a time-series statistical regression approach [18]. The method relies on the assumption that vegetation cover and terrain roughness parameters do not change too quickly between adjacent passes, and acquisition from the different sensors take place at the same time. Frequent acquisitions are a good way to approximately satisfy these constraints.

Unfortunately, due to a hardware malfunction, in July 2015 the SMAP radar discontinued its operation. The radiometer continued to operate normally, but without its matching radar acquisition the datasets produced were obviously incomplete. A solution to incorporate radar data was then proposed, which consisted of combining active microwave acquisitions from an independent mission with the stand-alone SMAP radiometer. An in-depth review of all possible active sources led to selecting the Copernicus Sentinel-1A/1B as the most suitable one [19]. The decision was largely driven by the mutual orbit configurations of the Sentinel-1 and SMAP satellites, which offer a minimal time difference between their respective acquisitions on any given area of interest. The fusion of SMAP and Sentinel-1 data generates a high-resolution soil moisture product (3 km), validating the feasibility of merging multifarious sources. Sentinel-1's SAR presents differences in configuration and characteristics from SMAP's, starting from the frequency of operation, which is C-band for the former and L-band for the latter.

The promising results of this early work proved that the active and passive components of the system need not operate in the same microwave band. In addition to this,

if more sources of active data from different bands can be used, then another interesting development can be envisaged. In particular, a research question can be posed in terms of whether the use of data from multiple active sources affords either obtaining more accurate estimates of active–passive co-variation parameters for a given observation period, or shortening the minimum observation period by increasing the temporal density of active samples, for a given accuracy. In this framework, this paper addresses a preliminary comparison of fresh and past results obtained from C-, X-, and L-band active sensing data. Increasing availability of X-band data thanks to existing satellites and the forthcoming COSMO 2nd generation constellation suggests that the X-band could be considered for the active component, as has been done for the C-band. In this regard, the proposed study presents a first estimate of the covariation parameter concerning surface soil moisture attained by using X-band radars as the active microwave signal coupled with SMAP radiometer. The selected X-band sources are the German Earth-observation satellite TerraSAR-X [20] and the Italian COSMO-SkyMed [21]. The study was conducted upon two different areas identified as suitable for the analysis, located in Germany (agricultural site) and Brazil (woody dense vegetation site). The paper presents the context, the analysis, and discusses results.

## 2. Context and Models Used

In this work, we first considered the model described in [1], conceived to study soil moisture behavior as derived from concurrent active and passive microwave signals. The covariation of the two signals, denoted as $\beta$, is retrieved through a single-pass estimation, meaning that a single pair of active and passive observations is used to calculate the parameter. Similar results have been obtained by applying a statistical approach based on time-series regressions [22], albeit at a lower spatial resolution due to the usage of coarser radiometer and radar data.

Considering a typical scenario consisting of a soil surface covered by a vegetation layer, for instance spontaneous vegetation, crop fields or forests, the incoming and outgoing electromagnetic radiation is comprised of a ground contribution (G), i.e., ground emission or scattering, a contribution consisting of the interactions between ground and vegetated cover (I), and, ultimately, a vegetation contribution of direct emission or scattering in the canopy (V). According to this scheme, both the emission and the active backscattered signal can be decomposed into the three aforementioned contributions.

In particular, the former can be broken down into the following expression:

$$T_{BP} = f_V e_G T_G + (f_V f_G r_p)e_V T_V + e_V T_V \tag{1}$$

while the latter can be broken down into:

$$|S_{PP}|^2 = f_S r_p + f_D r_p + |S_{PP}^V|^2 \tag{2}$$

The brightness temperature $T_{BP}$ [K] is the parameter supplied by the radiometer dataset that provides the emissivity of a radiating body, when divided by its actual temperature. As shown in Equation (1), it is composed of three additive terms. The first one describes the upwelling ground emission $e_G$ multiplied by the loss caused by vegetation attenuation $f_V$. $T_G$ and $T_V$ represent, respectively, the physical temperatures of ground and vegetated layers, which, at certain observation times (early morning) [23], can be assumed to be in thermodynamic equilibrium and, therefore, equal to the same temperature $T$. By grouping $T$ and dividing $T_{BP}$ for it, the emissivity $e$ is retrieved. The second term summarizes the reciprocal action of ground and vegetation: the vegetation radiometric emission, $e_V$, directed towards the ground, is reflected by the coarse surface, as represented by the factor $f_G r_p$, and, then, it undergoes the attenuation loss $f_V$ when passing through the vegetated volume again. Finally, the last term accounts for the pure vegetation emission, $e_V T_V$.

In a similar way, the active signal backscattered from the scene consists of the three components in Equation (2) representing the direct backscatter from the ground ($f_S r_p$), the

interaction between vegetation and ground ($f_D r_p$), and the direct vegetation contribution ($|S_{PP}^V|^2$). $f_S$ incorporates the incoherent surface scattering losses, while $f_D$ incorporates the coherent double-bounce scattering.

The key parameter linking active and passive microwave signals is the smooth ground reflectivity $r_p$. $r_p$ provides a quantitative measure of both backscattered radiation from the surface and upwelling soil emission. By isolating the reflectivity term from Equations (1) and (2), it becomes possible to unify the two equations into one, expressing the relationship between radar backscatter and radiometer brightness temperature in linear form, thus comprising of a slope and an intercept term. The slope term represents the covariation parameter $\beta$, which can be written as:

$$\beta = \frac{\frac{T_{BP}}{T} - (f_V + e_V)}{|S_{PP}|^2 - |S_{PP}^V|^2} \tag{3}$$

From (3), the covariation parameter is the ratio between emission and backscatter after being corrected from the direct vegetation contributions, $(f_V + e_V)$ and $|S_{PP}^V|^2$. In order to calculate $\beta$ by using radiometer and SAR observations, the vegetation correction terms are modeled according to suitable models for microwave emission and backscatter of vegetated soil [24,25]. The radar backscatter part in the denominator of Equation (3) is modeled considering the Born series truncated at the first-order term, comprehensive of distorted Born and Foldy–Lax approximations [26], therefore becoming:

$$|S_{PP}|^2 - |S_{PP}^V|^2 = |S_{PP}|^2 - \mu_{PQ}^{PP}|S_{PQ}|^2 \tag{4}$$

In the equation above, the co-polarized vegetation scattering component, $|S_{PP}^V|^2$, which represents the direct vegetation contribution to the total backscattered radiation, is expressed as a function of the cross-polarized backscatter, $|S_{PQ}|^2$, given its major role through the multiple-bounce scattering mechanism within vegetated volumes [27]. $\mu_{PQ}^{PP}$ is calculated by using concurrent co-polarized and cross-polarized backscatter measurements provided by the employed dataset and by estimating their statistical regression slope as follows: $\mu_{PQ}^{PP} = \frac{\partial |S_{PP}|^2}{\partial |S_{PQ}|^2}$.

The parameters needed in order to perform the calculation are recovered from the radiometer dataset and the radar dataset contributed by the selected satellite mission. The possibility to combine different microwave wavelengths is the main advantage of the covariation model just described, other than its single-pass nature, allowing to expand the original L-band implementation to heterogeneous couples of active–passive instruments and providing the premises for our work.

In the considered period, however, very little cross-pol data were available from any of the selected X-band radars. Cross-pol data are needed to compute the parameter $\mu_{PQ}^{PP}$, and thus it was necessary to get around such a data availability issue. A simplified formulation was proposed in the past [28], relying on co-pol data only. Since the brightness temperature and the radar backscatter are negatively correlated when soil moisture variations occur, their relationship can be expressed in a more compact form as a linear expression, as shown in Equation (5), where $\beta$ represents the slope of the regression:

$$T_{BP} = \alpha + \beta \sigma_{PP} \tag{5}$$

$\alpha$ [K] is the intercept term of the linear equation. It depends on the intrinsic characteristics of the observed scene, in particular vegetation coverage type and soil roughness properties. In practice, the estimated values of $T_{BP}$ and $\sigma_{PP}$ are placed in a graph and their best fitting line is determined through a least squares approach. Then, $\alpha$ and $\beta$ are identified as the intercept and slope coefficients, respectively, of such a line. In our analysis, both of the formulations presented above have been assessed. The two equations return covariation estimates that are not directly comparable to each other, given their different

units of measurement; yet, they basically represent the same information in two different ways. As will be illustrated in Section 4.2, we have analyzed the relationship between the parameters computed considering and ignoring cross-polarimetric data, and found they strongly correlate, which encouraged us to relieve the requirement on availability of cross-pol data.

## 3. Study Sites and Data

### 3.1. Study Sites

The purpose of our work is the combination of multiple sources of passive and active microwave sensing data, in order to assess feasibility and potential of extending the analysis of active–passive mutual behavior to X-band data, in addition to the previously established results achieved with L- and C-bands. The study was conducted over two test sites, selected after an in-depth search based on a set of specific requirements.

Within the rather large SAR tiles on the earth surface and the even larger radiometer footprint of the used dataset, a selection of specific areas has been carried out. The definition of the regions of interest within overlapping stripes was done with a view to ensuring the vegetation cover within each single area (much smaller than the radiometer pixel, but larger than the radar pixel) was homogeneous, i.e., crop field or forest. It was important to define study regions that belonged simultaneously to a single radiometer footprint and to its overlaid radar strips to avoid introducing spurious variations. This is necessary to ensure fair comparison with the covariation parameter recovered from the SMAP/SMAP and the SMAP/Sentinel-1 datasets. In the end, three different areas were defined for Germany and two for Brazil by selecting distinct vegetation types and multiple couples of radiometer footprints and radar strips. The regions will be described in more detail when discussing the results achieved. The first constraint was the temporal period of interest. Global combinations of SMAP L-band radar and radiometer data, which we intended to use as a reference, are available only over a limited time interval. The latter corresponds to the life span of SMAP radar before its permanent failure, i.e., from 13 April 2015 to 7 July 2015. The second constraint is that the utilized dataset has to be polarimetric. In fact, according to Equation (3), both co-polarized and cross-polarized radar backscatter parameters are necessary to estimate the covariation value. This has made the search process particularly challenging, since typically cross-polarized backscatter data are globally very scarce. In fact, as already mentioned in Section 2, this requirement could not be satisfied in full, i.e., on a sufficient number of time samples along the relevant time interval. As a consequence, we had to scale back to a simplified $\beta$ computation (5); however, the decision on the areas was already taken based on the few polarimetric pieces of data available. Finally, the focus on mildly vegetated areas further restricted the options. In the end, our selection of sites based on TerraSAR-X and COSMO-SkyMed availability fell onto the following areas:

1.  An agricultural region in southern Germany around coordinates 47°58′12.0″N, 11°55′45.0″E (see Figure 1 for reference);
2.  A partly de-forested region in northern Brazil around coordinates 6°48′05.63″S, 55°24′46.22″W (see Figure 2 for reference).

### 3.2. Data

The datasets used in our work are listed in Table 1: L-band SMAP active/passive products to recover the passive components of the covariation equation and to retrieve reliable $\beta$ estimates suitable for performing a comparison; C-band Sentinel-1 data; and, the base for our contribution to the analysis, X-band TerraSAR-X and COSMO-SkyMed radar data.

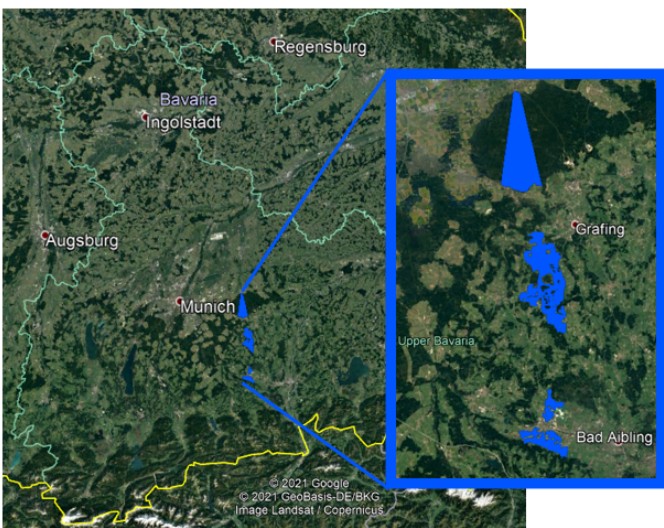

**Figure 1.** Picture showing the analyzed zone in RGB colors, regarding the German dataset in the Bavaria region, captured in GoogleEarth.

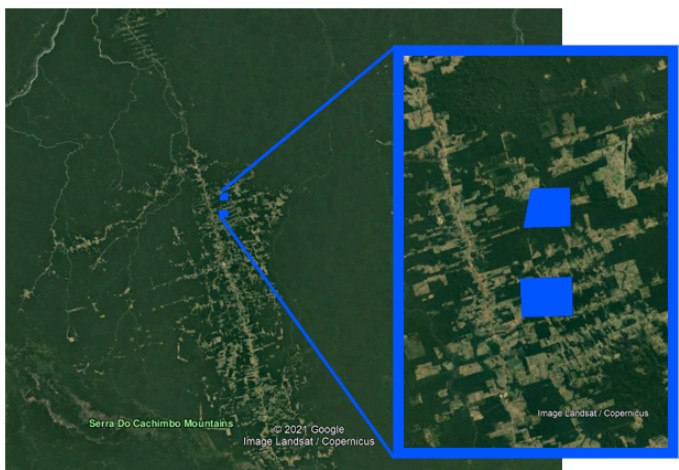

**Figure 2.** Picture showing the analyzed zone in RGB colors, regarding the Brazilian dataset in the State of Pará, captured in GoogleEarth.

**Table 1.** Overview of the datasets employed in our analysis.

| Mission (Sensor) | Dataset | Band (Polarization) | Spatial Posting | Temporal Resolution |
|---|---|---|---|---|
| SMAP (Radiometer) | SMAP Enhanced L2 Radiometer Half-Orbit 9 km EASE-Grid Soil Moisture, Version 3 | L-band | 9 km | 2–3 days |
| SMAP (radiometer/radar) | SMAP L2 Radiometer/Radar Half-Orbit 9 km EASE-Grid Soil Moisture, Version 3 | L-band | 9 km | 2–3 days |
| Sentinel-1 (radar) | SAR Standard L1 Product, GRD type | C-band (VV, VH, HH, HV) | 9 m | <3 days |
| TerraSAR-X (radar) | TSX-1.SAR.L1b-Stripmap | X-band (VV, VH) | 3 m | 11 days |
| COSMO-SkyMed (radar) | COSMO_SkyMed StripMap HIMAGE mode | X-band (VV, HH) | 5 m | 16 days |

The SMAP mission in its initial, fully operational configuration used to generate 24 data products, spanning a total of four levels of data processing. We selected Level-2 (L2) products; the Enhanced L2 Radiometer dataset contains calibrated and geolocated brightness temperatures at 9 km posting acquired by the SMAP radiometer during 6:00 a.m. descending and 6:00 p.m. ascending half-orbit passes. The L2 Radiometer/Radar dataset contains a combination of active and passive products and it provides covariation estimates at 9 km posting. The grid topology is EASE (equal-area scalable Earth) grid radiometer brightness temperature (36 km) with the EASE grid radar backscatter cross-section (3 km) [29].

Regarding Sentinel-1 radar data, level-1 ground range detected (GRD) products have been considered. They consist of SAR data projected from slant to ground range using the Earth ellipsoid model WGS84. The projection was corrected using the terrain height, increasing georeferencing precision. At the end, approximately square-shaped pixels were obtained, with reduced speckle, at the expense of a coarser spatial resolution (9 m) [30]. Moreover, it was possible to retrieve polarimetric data. On the same level of processing, we selected TerraSAR-X and COSMO-SkyMed products projected onto the ellipsoid model WGS84 and corrected from the spatial distortions caused by the different terrain heights through the use of Digital Elevation Models ($DEM_S$).

## 4. Data Processing and Analysis

### 4.1. Preparation of Radar Data

Before their usage in covariation estimates, radar data need suitable pre-processing and calibration. In our work, pre-processing consisted of the following steps: reprojection and cropping of the dataset, calibration, and local mean filtering. The cropping operation involved cutting out the selected area of interest from the received radar dataset, opportunely projected onto the same coordinate system (WGS 84 EPSG 4326). The reason for this procedure is to ensure spatial consistency among the different datasets used. To reduce the impact of bright reflective surfaces such as buildings, and also to reduce the impact of speckle noise, a mean filter was applied to the cropped data. Finally, a radiometric calibration step was implemented (in QGIS using dedicated plug-ins) in order to derive the actual backscatter values from the image pixel intensities (or digital numbers). Each mission features its own, specific radiometric calibration process and calibration factors as specified in the technical documents [30–32] and in the attached metadata products. In the case of TerraSAR-X, for example, radiometric calibration produces the $\sigma^0$ or *sigma naught* parameter, the radar reflectivity per unit area in ground range, through the formula [32]:

$$\sigma^0 = \beta_0 \cdot sin(\theta_{loc}) \tag{6}$$

This formula requires information on radar brightness ($\beta_0$) and the local incidence angle ($\theta_{loc}$) of the sample being calibrated. $\beta_0$ is obtained by multiplying the second power of the pixel values (DN) times the calibration factor ($K_s$):

$$\beta_0 = K_s \cdot |DN|^2 \tag{7}$$

$K_s$ values are passed in the form of XML files, attached as a satellite data support file; there exist specific calibration factors for co-polarized and for cross-polarized data. The local incidence angle is the angle between the radar beam and the normal to the illuminated surface and the corresponding file can be ordered optionally. In our case we decided to use the central incidence angle in order to simplify the procedure. A more accurate radiometric calibration requires detailed knowledge of $\theta_{loc}$. The other types of radar data used require analogous procedures, which we implemented in full using QGIS, specialized plugins and developing tailored scripts when needed. The interested reader is referred to the above-cited technical documents for more detailed information.

### 4.2. Preliminary Analysis: Full vs. Simplified Model

Our preliminary analysis concerned the implementation of Equation (5) by adopting both the co- and cross-polarization first, and then co-pol only. Results were compared with those sourced from NASA's archived and freely distributed computations of $\beta$, which we considered as a reference. In order to focus on the similarity in information content, we used mutual correlation to compare the data series. As can be seen from Table 2, the correlation factor between the covariation estimates computed considering both co- and cross-polarizations and those computed considering only co-polarizations was almost 1, meaning that cross-polarization does not impact significantly on the $\beta$ estimation process when X-band data are used. In the X-band, penetration into vegetation is much weaker and this may have a role in making vegetation-activated cross-pol less relevant. Therefore, in this study on the X-band, we focussed on covariation estimates using co-polarization only. This simplified the problem and conveys the additional advantage of expanding the choice of test areas, as rare X-band cross-pol data were no longer necessary. In fact, given the recorded cross-pol data availability in the concerned timeframe this decision was unavoidable if experiments were to be completed.

**Table 2.** Active–passive microwave covariation estimates regarding the first agricultural German site, computed with the SMAP radiometer plus TerraSAR-X radar combination. $\beta_{VH}^{VV}$ were computed considering both co- and cross-polarizations; $\beta^{VV}$ were computed considering only co-polarizations.

| Sensing Date (yyyy/MM/dd) | $\beta_{VH}^{VV}$ [K/dB] | $\beta^{VV}$ [K/dB] | Correlation Factor |
|---|---|---|---|
| 2015/06/11 | −3.97367 | −4.14314 | |
| 2015/06/19 | −3.10215 | −3.24648 | |
| 2015/06/22 | −3.65805 | −3.86859 | 0.99907 |
| 2015/06/30 | −3.98658 | −4.14102 | |
| 2015/07/03 | −4.80281 | −5.04388 | |

### 4.3. Covariation Analysis

As already mentioned in Section 3.1, constraints on the time period made it challenging to retrieve suitable polarimetric radar products: for instance, there were no cross-pol COSMO-SkyMed products at all in the selected areas of interest within the considered timeframe. Nevertheless, as discussed in Section 4.2, we were able to get around this problem using Equation (5) to estimate $\beta$, which does not require the cross-polarized backscatter parameter. The extremely high correlation between the results achieved with Equation (3) and those achieved with Equation (5) confirmed that turning down the cross-pol data and using the simplified model did not seriously impair results. A plausible reason can be sought in the different penetration into the target that comes with the change in the operational frequency of the instrument: at the X-band, the penetration of the emitted radiation through the vegetation cover is smaller than at the C-band. Since the cross-polarized backscatter is directly related to the volume scattering caused by the presence of a vegetation layer over the surface, the impact of the parameter tends to become negligible with smaller-wavelength signals unable to infiltrate into dense vegetation volumes. Therefore, it was deemed reasonable to disregard the cross-polarized radar backscatter in exchange for access to a broader choice of test cases.

## 5. Results and Discussion

As previously described, three regions were selected on the German test site: two crop fields and a forest. The two agricultural regions belong to two different radiometer footprints projected onto the Earth surface, thus building a broader pool of passive sensing samples and preparing a more solid analysis of the covariation estimates. As already mentioned, our results were produced using Equation (5), and are represented in the graphs in Figures 3–5. On their vertical axes, the two corresponding covariation estimates are traced, i.e., ours and the reference NASA estimate. The analysis was conducted on all of

the previously mentioned instruments, i.e., Sentinel-1, TerraSAR-X, and COSMO-SkyMed, for every selected test site. For the sake of conciseness, only the most significant plots are reported in this paper.

While results were calculated with both the vertical and horizontal polarization, only the vertical polarization will be shown due to the similarity in outcomes.

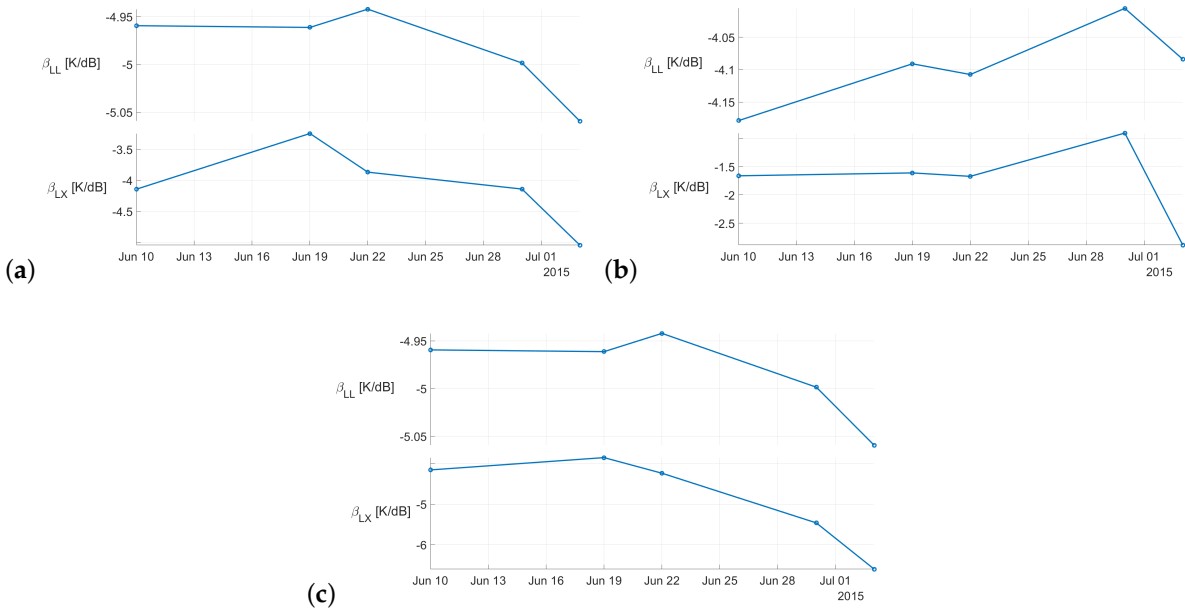

**Figure 3.** Active–passive microwave covariation estimates for the German test site: (**a**) crop field n. 1; (**b**) crop field n. 2; and (**c**) forest site. $\beta^{LL}$ refers to the SMAP radiometer plus SMAP radar combination; $\beta^{LX}$ refers to the SMAP radiometer plus TerraSAR-X radar combination. Acquisition times were all located around 6 a.m. local time.

Visual analysis of the graphs led us to observe that:

- Absolute values showed systematic differences;
- Variations, however, in general appeared to be correlated.

Systematic differences in absolute values may be due to the different radar reflectivity of the same structures at the different wavelengths of the systems involved. We decided to further investigate the correlation between the two series, in order to determine whether the information contribution from X-band data is consistent with information conveyed by L-band data when active–passive covariation is concerned. We aimed at understanding whether X-band data may add relevant clues or rather pour noisy data into the process. We thus computed and analyzed the correlation factor associated with the two covariation sequences, one from the original data from NASA and the other from our mixed L+X-band data. Correlation results are shown in Table 3.

The magnitude of $\beta$ expresses the sensitivity of L-band radiometer signatures to changes in L-band, C-band, and X-band radar signatures. By observing the high correlation values between the reference covariation parameters provided by NASA ($\beta^{LL}$) and the ones estimated by means of alternative active sources ($\beta^{LC}$, $\beta^{LX}$), it appears that changing the operational band of the active microwave signal did not result in substantial modifications to the overall behavior of the sensed variables. Further analyzing the results, one can notice that the correlation factor for the second German crop field shows an unexpectedly small value in comparison with the one calculated for the first field, despite featuring basically the same land cover type. However, the recurrence of a low correlation across all combinations of active sources suggests that the error may reside in the radiometric datum acquired on the specific footprint under test, possibly due to local radio frequency interference (RFI), a well-known problem for passive microwave sensing [31].

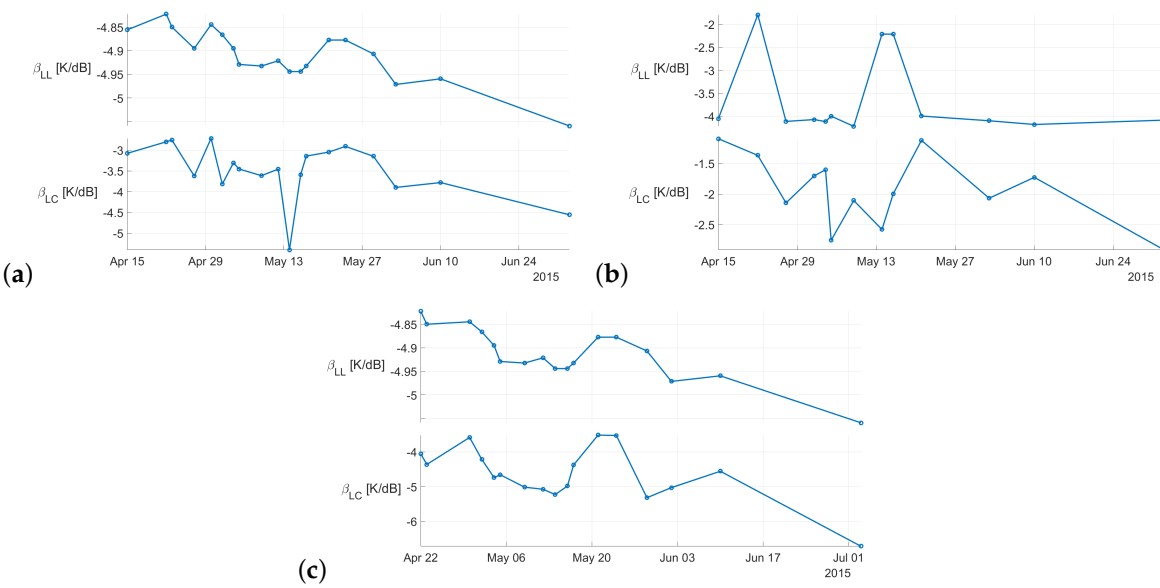

**Figure 4.** Active–passive microwave covariation estimates for the German test site: (**a**) crop field n. 1; (**b**) crop field n. 2; and (**c**) forest site. $\beta^{LL}$ refers to the SMAP radiometer plus SMAP radar combination; $\beta^{LC}$ refers to the SMAP radiometer plus Sentinel-1 radar combination.

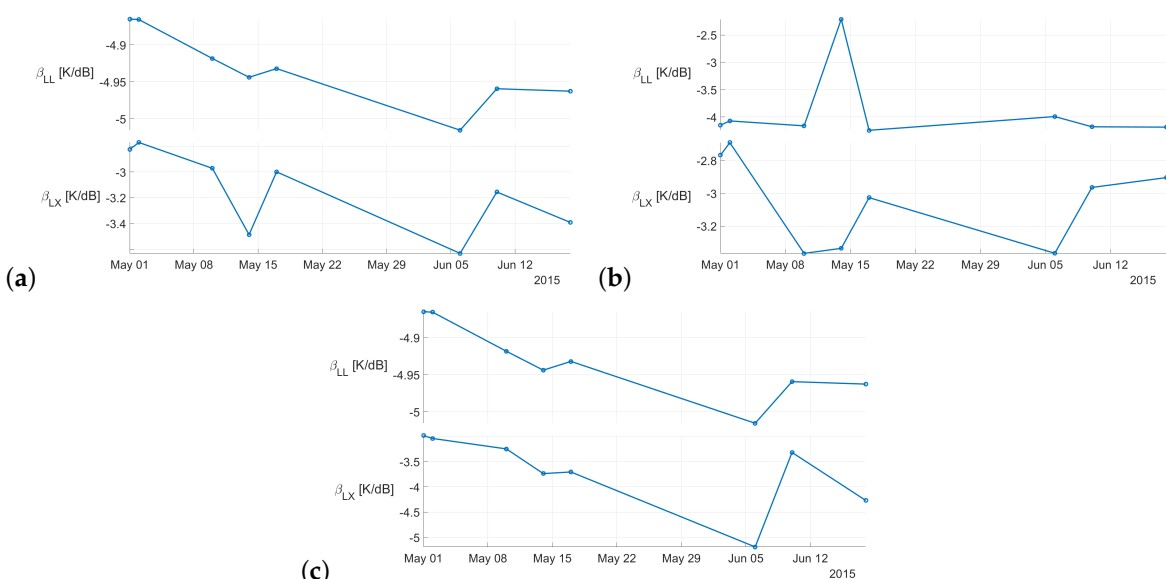

**Figure 5.** Active–passive microwave covariation estimates for the German test site: (**a**) crop field n. 1; (**b**) crop field n. 2; and (**c**) forest site. $\beta^{LL}$ refers to the SMAP radiometer plus SMAP radar combination; $\beta^{LX}$ refers to the SMAP radiometer plus Cosmo-SkyMed radar combination.

If one excludes the (likely RFI corrupted) results for the "Crop field #2" test site, the correlation factors for X-band data always exceed 0.82. This suggests a good agreement between the L-band and X-band active data contributions, which is even greater than with C-band active data, whose correlation averaged around 0.76 on the two remaining test sites.

**Table 3.** Active–passive microwave covariation estimates regarding the German sites. $\beta^{LL}$ refers to the SMAP radiometer plus SMAP radar combination; $\beta^{LX}$ refers to the SMAP radiometer plus TerraSAR-X (TSX) or COSMO-SkyMed (C/S) radar combination; $\beta^{LC}$ refers to the SMAP radiometer plus Sentinel-1 radar combination.

| Test Site | Covariation Series Computed on | Corr. Factor between the Two Considered Series |
|---|---|---|
| Crop field n.1 | $\beta^{LL}$ and $\beta^{LX}$ (TSX) | 0.82328 |
| Crop field n.2 | $\beta^{LL}$ and $\beta^{LX}$ (TSX) | 0.31269 |
| Forest | $\beta^{LL}$ and $\beta^{LX}$ (TSX) | 0.95762 |
| Crop field n.1 | $\beta^{LL}$ and $\beta^{LX}$ (C/S) | 0.89894 |
| Crop field n.2 | $\beta^{LL}$ and $\beta^{LX}$ (C/S) | −0.44025 |
| Forest | $\beta^{LL}$ and $\beta^{LX}$ (C/S) | 0.87690 |
| Crop field n.1 | $\beta^{LL}$ and $\beta^{LC}$ | 0.70064 |
| Crop field n.2 | $\beta^{LL}$ and $\beta^{LC}$ | 0.00305 |
| Forest | $\beta^{LL}$ and $\beta^{LC}$ | 0.82348 |

The second test area is a partly deforested Amazon area in NE Brazil, where two fields have been identified with similar, mixed land cover including rainforest and low vegetation. No suitable COSMO-SkyMed data were found in the reference timeframe either, so the experiment was conducted on TerraSAR-X data only. Similarly, no Sentinel-1 data with the same features as the previous experiments could be retrieved in the targeted time frame, so a comparison with C-band data could not be performed. Numerical results are reported in Table 4. The outcome of the analysis on the Brazilian test area seems to support the conclusions from the German test area. In particular:

- The difference in average $\beta$ values from L-band and from X-band data was nearly constant across the two test fields, i.e., the same systematic displacement was observed between the L-based and X-based estimation of $\beta$;
- Correlation factors between L-based and X-based $\beta$ values were high to extremely high.

These data are naturally to be taken with caution, as the statistical basis is very narrow, due to the limiting factors discussed in Section 2. Having warned the reader about this, however, some interesting similarities may be noticed between results from the German and the Brazilian test areas. One interesting point to note is that the correlation holds also on land cover types such as forest, where backscatter mechanisms for active X-band data are expected to be substantially different from those for L-band data [32]. This might reflect the fact that forests are more homogeneous in their interaction with microwaves with respect to agricultural areas, which may include plants with different heights and shapes.

**Table 4.** Active–passive microwave covariation estimates regarding the Brazilian sites. $\beta^{LL}$ refers to the SMAP radiometer plus SMAP radar combination; $\beta^{LX}$ refers to the SMAP radiometer plus TerraSAR-X radar combination.

| Test Site | Sensing Date | $\beta^{LL}$ [K/dB] | $\beta^{LX}$ [K/dB] | Correlation Factor |
|---|---|---|---|---|
| | 2015/05/04 | −3.83956 | −5.57806 | |
| Field n. 1 | 2015/05/26 | −3.85407 | −5.84327 | 0.99955 |
| | 2015/06/17 | −3.84065 | −5.60636 | |
| | 2015/05/04 | −3.61885 | −5.07345 | |
| Field n. 2 | 2015/05/26 | −3.63109 | −5.53964 | 0.76329 |
| | 2015/06/17 | −3.61960 | −5.40265 | |

## 6. Conclusions

In this paper, we have presented an analysis of the features of estimated active–passive covariation when applied to X-band data rather than to the traditionally used L-band and C-band data. The aim of our investigation was to asses whether X-band radar data, which are now becoming increasingly available, are suitable to contribute to

mapping of land cover parameters such as soil moisture or vegetation water content in the framework of an active–passive covariation analysis. This is important in order to assess whether incorporating data from multiple missions in the covariation analysis leads to improvements in parameter retrieval. The question is not trivial, as X-band waves penetrate much less into vegetation than C-band and L-band waves; yet a significant contribution from the ground may still be collected. This is especially true in the case of a shallow layer of vegetation as was characteristic of some of the considered test sites. Our analysis revealed very high degrees of correlation between the $\beta$ values computed using X-band, L-band, and C-band data (around 0.7 to 0.8 as seen in Tables 3 and 4), suggesting that X-band active data can contribute to the estimation of co-variation, effectively taking the place of other C-band elements in a time series; this should enable computation of $\beta$ on shorter time series. Only one exception was observed, which was attributed to local disturbance factors because of the complete lack of connection of its results with any others. The correlation of results across different radar bands, however, comes with a significant offset in estimated values, which probably depends on the different typical responses of vegetated land cover at the different wavelengths considered. This offset needs to be determined and removed to make the results usable for the intended purpose. Unfortunately, the need to use original SMAP data as a reference proved a limiting constraint that resulted in scarce availability of suitable X-band data from both TerraSAR-X and COSMO-SkyMed sources. This means that the hypothesis should be further tested to prove its robustness in a more general case. Given the extreme scarcity of suitable spaceborne test data, however, this may only be achieved by leveraging other independent sources for the reference $\beta$ value on the test sites. Another key issue that must be mentioned here is the availability of X-band SAR data, which has significantly constrained our capacity to deepen the investigation. Currently, there exist very few X-band spaceborne missions run by national space agencies. New actors from the private sector are, however, entering into business, such as ICEYE [33] and Capella Space [34], who offer high spatial resolution and frequent revisit times, such that an almost perfect time agreement can be achieved among the passive and the active sensing operations. Despite their data not being distributed for free, possible future commercial services for vegetation and soil moisture monitoring will be in a position to rely on abundant and ubiquitous X-band data sources.

Future directions of investigation include:

- Comparing X-band-derived estimates and the standard estimates provided by NASA using a combination of passive SMAP and Sentinel-1 data, on a geographically broader set of test areas;
- Incorporating active L-band data from the JAXA sensor ALOS.

The long-term goal of our research is to devise an algorithm for co-variation computation that does not require long time series of C-band data, because part of the required information is provided by active X-band and active L-band data acquired at similar times to the other contributions.

**Author Contributions:** Conceptualization, F.D., D.E.; methodology, F.D., E.A.; software, E.A., S.B.; investigation, validation, formal analysis, E.A., S.B. and F.D.; data curation, E.A., S.B.; writing—original draft preparation, E.A. and S.B.; writing—review and editing, F.D., D.E.; supervision, F.D., D.E.; funding acquisition, F.D. All authors have read and agreed to the published version of the manuscript.

**Funding:** This research was partly funded by the University of Pavia through its internal "Boston incoming" project, and partly by the European Commission through the MSCA-RISE project "EOX-POSURE", Grant Agreement no. 734541.

**Institutional Review Board Statement:** Not applicable.

**Informed Consent Statement:** Not applicable.

**Data Availability Statement:** Sentinel-1 data are available at the Copernicus Open Access Hub. COSMO/SkyMed data are available by applying to the Open Call by the Italian Space Agency, details

available at https://www.asi.it/bandi_e_concorsi/open-call-per-la-comunita-scientifica-utilizzo-dei-dati-cosmo-skymed-prima-e-seconda-generazione/ (accessed on 18 March 2021). TerraSAR-X data are available by applying to the TerraSAR-X Science Service System, details available at https://www.sss.terrasar-x.dlr.de/ (accessed on 18 March 2021). SMAP data are available through the National Snow and Ice Data Center portal.

**Acknowledgments:** The authors wish to acknowledge the Italian Space Agency (ASI) and the German Space Agency (DLR) for providing X-band satellite data for free for scientific use through their respective Announcement of Opportunity (AO) schemes. The authors also wish to thank Narendra N. Das, at NASA-JPL at the time of this research work, for his support with technicalities of SMAP data representation.

**Conflicts of Interest:** The authors declare no conflict of interest.

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
