# Peer review of "Covariation of Passive–Active Microwave Measurements over Vegetated Surfaces: Case Studies at L-Band Passive and L-, C- and X-Band Active"

_remotesensing, doi:10.3390/rs13091786_

Round 1
Reviewer 1 Report
The paper has been extensively reworked, and I think that the overall readability as well as the presentation and discussion of the results has improved quite a bit.
The new figures allow a much easier visual inspection and the reasons for your choices on used equations and study-sites are now much clearer.
While the data-scarcity is still the primary concern for this study,
I aggree that this is something that can be improved in subsequent studies.
Alltogether I think the paper is now ready for publication.
(just some very minor comments below)
Comments:
84 : "This suggests that band X could be used... ... as it has been done for band C"
I would suggest using the terms "X-band" and "C-band"
212 : "The second "constraint is the utilized dataset has to be polarimetric"
The second constraint is, that the utilized dataset has to be polarimetric
215 : "Actually, as explained in ..., this requirement "
I guess you missed putting in the reference ?
217 : "As a consequence, we had to scale back to a simplified β computation; however, the decision..."
I'd suggest referring to the equation as well... e.g. something like:
"... we had scale back to the simplified β computation (4); however..."
332 "...whose correlation averages around 0.76 on the surviving test sites."
...on the remaining test-sites...
Reviewer 2 Report
The authors are referred to the attached file.

Round 2
Reviewer 2 Report
Thank you for your answers and clarifications, the manuscript has improved considerably.
I still wonder about the SAR preprocessing, which was performed in QGIS. Are these self-developed or publicly available plugins? It would be relevant to mention the names and possibly references.